# Analysis of Mutational Status of *IGHV*, and Cytokine Polymorphisms as Prognostic Factors in Chronic Lymphocytic Leukemia: The Romanian Experience

**DOI:** 10.3390/ijms25031799

**Published:** 2024-02-01

**Authors:** Beata Balla, Florin Tripon, Erzsebet Lazar, Claudia Bănescu

**Affiliations:** 1Department of Medical Genetics, George Emil Palade University of Medicine, Pharmacy, Science and Technology of Targu Mures, 540139 Targu Mures, Romania; beakardos@gmail.com (B.B.); claudia.banescu@gmail.com (C.B.); 2Center for Advanced Medical and Pharmaceutical Research, Genetics Laboratory, George Emil Palade University of Medicine, Pharmacy, Science and Technology of Targu Mures, 540139 Targu Mures, Romania; 3Medical Genetics Laboratory, Emergency County Hospital of Targu Mures, 540136 Targu Mures, Romania; 4Department of Internal Medicine, George Emil Palade University of Medicine, Pharmacy, Science and Technology of Targu Mures, 540139 Targu Mures, Romania; erzsebetlazarbenedek@gmail.com

**Keywords:** *IGHV*, cytokines, prognosis, CLL, *IL-10*, *TNF-α*

## Abstract

The aim of the current study was to assess the associations between genetic risk factors (such as the mutational status of the *IGHV* gene and polymorphisms of the *IL-10* and *TNF-α* genes) and CLL risk, prognosis, and overall survival. Another goal of this study was to evaluate the multivariate effect of the combination of multiple genetic risk factors (mutational status of the *IGHV* gene, somatic mutations, DNA CNVs, and cytokine SNPs) on the clinical characteristics and survival of patients. A total of 125 CLL patients and 239 healthy controls were included for comparative SNP analysis. *IL-10* (rs1800896 and rs1800872) and *TNF-α* (rs361525 and rs1800750) SNPs and haplotypes were not associated with CLL risk. The absence of hypermutation in the *IGHV* gene was shown to be of important prognostic value, being associated with short OS. Further individual risk factors for short OS were an age above 65 years at diagnosis and the presence of somatic mutations and/or CNVs. In our multivariable analysis, the presence of somatic mutations and the *IL-10* rs1800872 variant allele, and the association of CNVs with the *IL-10* rs1800896 variant allele, were identified as risk factors for short OS. Moreover, the OS in unmutated *IGHV* patients was additionally affected (decreased) by the presence of CNVs and/or somatic mutations. Similarly, *IL-10* rs1800896 modulated the OS in unmutated *IGHV* patients with CNVs.

## 1. Introduction

Chronic lymphocytic leukemia (CLL) holds its position as the most prevalent form of leukemia among adults in the Western world, constituting approximately 25% of adult leukemia cases. From a clinical perspective, patients with CLL often exhibit asymptomatic elevation of lymphocytes or leukocytes in the peripheral blood (predominantly lymphocytosis), along with manifestations such as lymphadenopathy, hepatosplenomegaly, bone marrow dysfunction, recurrent infections, and occasionally, autoimmune hemolytic anemia or autoimmune thrombocytopenia [1]. CLL displays a remarkable degree of heterogeneity, with certain patients never needing treatment, while others experience rapid disease progression despite intensive therapy. Genetic abnormalities, such as chromosomal aberrations (copy number variations—CNVs), gene mutations, and single-nucleotide polymorphisms (SNP), are very common in CLL, and represent important prognostic risk factors. The mutational status of the variable region of the immunoglobulin heavy-chain (*IGHV*) gene has also emerged as a significant prognostic marker. It plays a crucial role in distinguishing variations in disease outcomes and is progressively becoming influential in guiding treatment decisions [1,2,3,4]. The presence of SNPs in cytokine genes indicates that specific alleles may contribute to variations in cytokine production. This suggests that genetic variants in cytokine genes could potentially influence gene expression and plasma levels, and, consequently, play a role in the development of hematological malignancies [5].

### 1.1. IGHV Gene Hypermutation in CLL

*IGH* genes encode the variable (V) regions responsible for antigen binding and the constant (C) regions specific to the different isotypes of immunoglobulin heavy-chain proteins (IGHV). By combining various segments of these genes, a vast array of sequences are generated, enabling the efficient recognition of diverse antigens [4,6].

Somatic hypermutation (SHM) is a natural biological process that takes place in B cells as they migrate through lymph nodes. This process takes place in antigen-stimulated germinal center B cells and plays an important role in enhancing the efficiency of cells involved in the diverse immune response [4,7,8]. The transformation into malignancy can occur in B cells either before or after the SHM process in the genes responsible for encoding the IGHV region of the leukemic B cell receptor (BCR) [7]. Consequently, in the case of CLL, two distinct types of B-cell clones can be identified based on variations in the mutational status of the *IGHV* gene compared to the germline sequence. Patients with mutated CLL (M-CLL) refer to individuals whose CLL clone exhibits a mutation of 2% or more from the reference germline sequence in the *IGHV* region. On the other hand, patients with unmutated *IGHV* (U-CLL) clones are characterized by sequences that have mutations of less than 2% from the germline sequence [9,10].

The status of SHM serves as a significant prognostic marker in CLL, offering valuable insights into disease outcomes. The absence of *IGHV* mutation has been linked to an unfavorable prognosis [4]. Specifically, in the context of chemoimmunotherapy, the presence of unmutated *IGHV* genes has been identified as a predictor of shorter progression-free survival [11]. Some theories suggest that the connection between a less aggressive form of the disease and the presence of mutated *IGHV* sequences may be attributed to a reduction in autoreactivity in the membrane immunoglobulin (IG). IG serves as a crucial element of the BCR. In this context, it is believed that the diminished autoreactivity restricts the BCR’s capability to transmit trophic signals to leukemic B cells, thereby influencing the disease’s aggressiveness [12]. Consequently, SHM testing is recommended in CLL, enabling accurate assessment and prognostication of the disease [4].

### 1.2. Cytokines in CLL

Cytokines, including interleukins (ILs), growth factors, and interferons, have a significant impact on regulating the inflammatory response and are implicated in the development of cancer. Chronic inflammation is closely linked to the release of diverse mediators, including pro-inflammatory and oncogenic substances such as reactive nitrogen oxygen species, inflammatory cytokines (such as IL-1β, IL-2, IL-6, and tumor necrosis factor alpha [TNF-α]), growth factors, and chemokines [13]. Cytokines are classified into pro-inflammatory (such as IL-6, IL-17, IL-18, TNF-α, interferon gamma (IFN-γ), etc.) and anti-inflammatory (such as IL-4, IL-10, IL-13, etc.) based on their inflammatory activity [13,14,15,16,17].

With increasing emphasis on immunity in the exploration of cancer etiology, more and more research is focusing on this area. As an immunosuppressive molecule that enables tumors to evade immune surveillance, IL-10 might act as a potential promoter of tumor growth, leading to more aggressive behavior of malignant cells. Conversely, due to its immune-stimulating and anti-angiogenic properties, IL-10 is believed to have the ability to prevent or reduce tumor growth and distant metastasis. It has been observed that the expression of IL-10 is either overexpressed or deficient under various pathophysiological conditions, depending on the specific cancers being analyzed [13,18,19,20,21,22,23,24]. Therefore, IL-10 is an immunoregulatory cytokine that exhibits multifunctional effects on tumor cells. It promotes the accumulation of certain malignant B-lymphocyte cells and plays a suppressive role in the immune response against tumor cell proliferation. Additionally, IL-10 has both stimulatory and suppressive functions in T lymphocytes. Some studies suggest that IL-10 suppresses cancer cells by stimulating CD8+ T cells, while other research indicates that IL-10 can lead to the stimulation of tumor cell proliferation [25,26,27,28,29,30,31,32].

CLL patients exhibit elevated levels of plasma IL-10 compared to healthy individuals. In laboratory settings, CLL cells derived from human subjects produce IL-10 when their BCRs are cross-linked [33]. The *IL-10* gene ranks within the top 1% of the most highly differentially methylated genes between unmutated CLL (U-CLL) and mutated CLL (M-CLL), being hypomethylated in M-CLL. Consequently, M-CLL produces more IL-10 than U-CLL following TLR receptor stimulation [26,34,35]. Moreover, IL-10 production in M-CLL cases has been correlated with reduced immune activity [26,27].

CLL cells exhibit varying levels of anergy, which is linked to the BCR and the *IGHV* regions, whether they are U-CLL or M-CLL. The degree of anergy is associated with lower levels of surface IgM and reduced signaling capacity, particularly noticeable in M-CLL. Additionally, there are differences in DNA methylation patterns between M-CLL and U-CLL. Patients from both subsets experience immunosuppression, even in the early stages, potentially influenced by the capacity of activated CLL cells to produce IL-10. These findings indicate that IL-10 production is regulated through epigenetic mechanisms. In cases of U-CLL, where IL-10 production is reduced, a higher tumor load might compensate for this decrease, contributing to the clinical immunosuppression observed in both CLL subsets [27].

The *IL-10* gene is composed of five exons, is located on chromosome 1, and has been mapped to the junction between 1q31 and 1q32.14. IL-10 plasma levels and transcriptional activity have been associated with three polymorphisms [−592C/A (rs1800872), −819C/T (rs1800871), and −1082A/G (rs1800896)] located in the promoter region of the *IL-10* gene [22,28,29], which influence gene transcription and protein production [17,30]. rs1800896 has been reported to be associated with a ≅32% reduction in mean IL-10 production in vitro [17,31].

Tumor necrosis factor (TNF) is a proinflammatory cytokine primarily synthesized by activated macrophages, natural killer cells, and T lymphocytes. It has multiple effects, including leukocyte trafficking and the clearance of immune complexes, but it also plays a crucial role in the development of rheumatoid arthritis. TNF has significant effects on bone remodeling and infection control. The release of TNF by macrophages appears to be essential for the formation and maintenance of granulomas and plays a critical role in defending against intracellular organisms. The precise role of TNF in cancer is not fully understood, but it seems to be associated with its varying concentrations at different stages of tumorigenesis. High levels of TNF are produced in the heart, and although the mechanisms by which TNF mediates cardiac injury are not entirely clear, they appear to depend on its concentrations. TNF also promotes dyslipidemia and insulin resistance, both of which are traditional risk factors for atherosclerotic processes. Taken together, it is evident that TNF is a pleiotropic cytokine involved in multiple homeostatic and pathological mechanisms [32].

The *TNF-α* gene is situated in the extensively polymorphic major histocompatibility complex (MHC) region on chromosome 6p21.3. Variations located in the promoter region of the *TNF-α* gene (such as rs1800750 at position −376G/A and rs361525 at position −238G/A) [36,37] are believed to impact the expression of *TNF-α* and are recognized as functional single-nucleotide polymorphisms (SNPs) [35]. The rs361525 polymorphism has been linked to different hemato-oncologic conditions such as chronic myeloid leukemia (CML) [13]; other types of cancers including breast cancer, lymphoma [37], gastric cancer [38], and lung cancer [39]; and various other medical conditions such as graft versus host disease, ear infections in children, nasal polyps, diabetes, the decline of verbal memory, and sudden infant death syndrome [37], as well as liver cirrhosis [40], psoriasis [41], and systemic sclerosis [42]. The rs1800750 polymorphism has also been investigated in different conditions, like cancer, diabetes, migraine, or multiple sclerosis [36], but to the best of our knowledge, not in CLL.

Elevated levels of IL-10 and TNF have been implicated in the pathophysiology of CLL and are associated with a poor prognosis [43].

The aim of the current study was to assess the associations between genetic risk factors (such as the mutational status of the *IGHV* gene and polymorphisms of the *IL-10* and *TNF-α* genes) and CLL risk, prognosis, and overall survival. Another goal of this study was to evaluate the multivariate effect of the combination of multiple genetic risk factors (mutational status of the *IGHV* gene, somatic mutations, CNVs, and cytokine SNPs) on the clinical characteristics and survival of patients.

## 2. Results

### 2.1. Description of CLL and Control Study Groups

Our study group comprised of 78 male and 47 female patients, with an average age of 64.9 years (min: 31, max: 87, SD: 11.22). The control group for SNP analysis included 239 healthy individuals with similar sex and age distribution.

Out of the 125 CLL patients, 84 (67.2%) patients had U-CLL, and 41 (32.8%) had M-CLL.

The associations of *IGHV* hypermutation with copy number variations (CNVs) and/or somatic mutations are presented in Table 1, where we observe a statistically significant association of CNVs with the absence of SHM (*p* = 0.009), as well as the combined association of CNVs and somatic mutations with the absence of SHM (*p* = 0.02). Furthermore, the association of *NOTCH1* mutation with the absence of SHM also shows a tendency toward statistical significance (*p* = 0.07).

### 2.2. Investigated SNPs and Risk of CLL

The distribution of genotypes for the investigated *TNF-α* and *IL-10* cytokine SNPs in the CLL patients and controls is shown in Table 2. There were no deviations from Hardy–Weinberg equilibrium (HWE) observed in either the CLL cases or controls for the *IL-10* rs1800896, rs1800872, or *TNF-α* rs361525 and rs1800750 genotypes.

As seen in Table 2, there were no significant differences in genotype distribution between the CLL and control groups for rs1800896, rs1800872, rs361525, and rs1800750 (*p* > 0.05).

Further, we analyzed the presence of different haplotypes in both the CLL and control groups. For the two loci in *IL-10*, out of the nine expected, only eight were present (Table 3a). The most frequent haplotypes were TCGG and TCGT (carrying a variant allele in one or both SNPs) in both the CLL and control groups. It was noticeable that the TCTT and CCGT haplotypes were only present in the control group. The haplotypes more prevalent in the CLL group were TTGT, TTTT, TCGG, and TCGT. The CCGG haplotype carrying a homozygous variant genotype in rs1800896 was more prevalent in the control group, while the TTTT carrying a homozygous variant genotype in rs1800872 was more prevalent in the CLL group. The TTGG haplotype carrying only wild-type alleles on both loci was more prevalent in the control group. For the two polymorphisms in the *TNF-α* gene, three out of the four expected haplotypes were present (Table 3b). We observed that the GAGA haplotype, carrying the variant allele in both loci, was present only in the CLL group. None of the haplotypes showed a statistically significant association with CLL.

No statistically significant differences were found (*p* > 0.05) in the analysis of the combined variant genotypes of two, three, of four variant genotypes in the two study groups either.

### 2.3. Associations between Studied Genetic Aberrations and Clinical Factors

Changes in the clinical characteristics of patients based on the presence of variant genotypes of the studied SNPs, as well as the absence of SHM alone or in association with CNVs and/or somatic mutations, are described in the Appendix A. When analyzing these changes, we observed that U-CLL was more prevalent in female patients (*p* = 0.02), and was associated with anemia in a significant manner when occurring as a single anomaly (*p* = 0.03), as well as when co-occuring with CNVs (*p* < 0.001). Further we observed a significant association of low platelet count (*p* < 0.001) and high WBC count (*p* = 0.004) in those with co-occurance of unmutated *IGHV* and CNVs. Variant genotypes of the *IL-10* and *TNF-α* SNPs did not have an influence over the clinical factors of CLL patients (*p* > 0.05).

### 2.4. The Impact of Genetic Aberrations and Clinical Factors on Overall Survival

Kaplan–Meier analysis showed that an age above 65 years had negative influence on the overall survival (OS) of patients (HR = 2.35, 95% CI = 1.21–4.72, *p* = 0.01), while other clinical characteristics (Appendix A) did not have such an influence on OS in our study group.

The OS observed in all M-CLL patients was 49.0 months, while in U-CLL patients, it was 38.4 months (21.6% less). Further, we continued investigating whether the survival of patients was influenced by the association of CNVs and/or somatic mutations with *IGHV* SHM (Figure 1), and we found that survival was negatively influenced by the presence of associated genetic aberrations compared to the absence of SHM alone (*p* = 0.02).

The studied variant genotypes did not have an impact on the OS of patients (*p* > 0.05), except for the two *TNF-α* SNPs; however, since it was only one patient carrying the variant A allele on both SNPs, we consider it to have no statistical significance. A lack of association was also observed for the combined variant genotypes (those with two, three, or four variant genotypes).

Bivariate cox regression analysis identified the association of somatic mutations and *IL-10* rs1800872 (HR = 5.01, 95% CI = 1.06–19.22, *p* = 0.02), and the association of CNVs and *IL-10* rs1800896 (HR = 2.3, 95% CI = 0.99–5.30, *p* = 0.04) as combined risk factors for short OS. Given the previously described various impacts of different CNVs [44] on OS, we analyzed the stratified impact of the four most frequently identified CNVs’ (del13q, del11q, del17p, and trisomy 12) co-occurrence with *IL-10* rs1800896. However, when taken separately, we did not find statistically significant associations, most likely due to the relatively small patient number in each group.

Finally, we summarized the concomitant presence of all investigated genetic risk factors in our CLL patients, available in the Appendix A. We observed that none of the patients tested negative for all investigated risk factors; a high majority of them (95.2%) had multiple genetic risk factors; and the most frequent co-occurrence was that of unmutated *IGHV* with a variant allele in the *IL-10* rs1800896 and CNVs. This specific combination proved to represent a combined risk factor for CLL (HR = 1.7, 95% CI = 0.7–3.9). Since the presence of the variant allele alone did not have an impact on OS, we wanted to analyze whether the addition of the variant allele to the other two aberrations had any impact. Therefore, we made a comparison between the unmutated *IGHV* + CNVs + wild-type allele in rs1800896 and the unmutated *IGHV* + CNVs + variant allele in rs1800896 (Figure 2), where the only difference between the two groups was the presence of the variant allele in rs1800896. A reduction in OS was noticeable, but with no statistical significance. The additive impact of multiple genetic aberrations should be analyzed in larger cohorts. A high degree of co-existence of unmutated *IGHV* and variant alleles of both *IL-10* SNPs was also noticeable (in 85.6% of patients), but without a statistically significant impact on OS (*p* > 0.05).

## 3. Discussion

To our knowledge, in Romania, patients with CLL are insufficiently or not at all investigated from a genetic point of view. Moreover, no previous studies have investigated the combined effect of all these genetic risk factors in CLL patients.

The absence of a somatic hypermutation in the *IGHV* gene has been linked to poor prognosis of CLL in several previously published studies [3,4,5,6,7,8,9,10], and our results are in accordance with those findings. Moreover, we found a prevalent association with CNVs, which may also contribute to the poor prognosis. Furthermore, our study showed that prognosis is strongly influenced by the association with CNVs and/or somatic mutations. The prognostic value of CNVs and somatic gene mutations in CLL has been previously reported by others [45,46,47,48] and also by our team [44]. The data of the current study showed similar results, since the presence of somatic mutations (HR = 2.4, 95% CI = 1.01–5.19, *p* = 0.03) and CNVs (HR = 2.1, 95% CI = 0.99–4.35, *p* = 0.05) were found to be individual risk factors for OS together with an age above 65 years at diagnosis.

The presence of anemia and thrombocytopenia observed in U-CLL patients was consistent with the Rai and Binet staging of patients, showing a more advanced stage of the disease, giving a further explanation of the poorer prognosis of these patients when multiple genetic aberrations were present. An age above 65 years at diagnosis proved to have a negative influence on the survival of our patients, which is consistent with other studies reporting similar impacts in hematological malignancies [13].

In a recent metanalysis by Gao S. et al., a strong association was found between rs1800896 and leukemia in non-Chinese patients, while the result was negative in Chinese patients [49]. In a study conducted by Aref S on a cohort of 50 CLL patients, they found that the variant allele of rs1800896 was significantly associated with advanced stages of CLL (*p* < 0.001), a short time to first treatment (*p* = 0.032), shorter progression-free survival (*p* = 0.044), and a lower response to treatment (*p* = 0.003) when compared to the wild genotype [23]. Furthermore, Gamaleldin M. et al. found that polymorphisms in the *IL-10* gene (CT and TT genotypes of rs1800896), as well as the presence of T alleles, could potentially elevate susceptibility to CLL development when compared to individuals with CC genotypes and C alleles in an Egyptian population. They concluded that the T allele might serve as a plausible risk factor for the disease [25]. In another study, Ovsepyanet et al. concluded that the TT genotype of rs1800896 was significantly linked with more advanced stages of CLL [50]. Basabaeen A. et al., on a Sudanese population, again found no association between CLL and rs1800896 [14]. As for rs1800872, in a metanalysis by Gao S., the authors found an association with leukemia risk in non-Chinese patients and no significant association in Chinese patients [49]. The cause of these discordances might lie in the different genetic makeup of the different populations. Our study, conducted on a Romanian population, found no statistically significant association of either rs1800896 or rs1800872 with CLL; however, the variant T allele of rs1800896 was found to represent an individual risk factor for CLL, similar to the result reported by Gamaleldin M. Moreover, it frequently co-existed with unmutated *IGHV* and CVNs, further representing combined risk factors for CLL.

The impact of *TNF-α* polymorphisms has been explored across various cancer types. While a link between cancer risk and *TNF-α* rs361525 polymorphisms has been established, the relationship remains contentious, as conflicting outcomes have been reported across diverse studies [13,42,51,52,53,54,55]. A previous study performed by Qidwai, T. et al. suggested that individuals carrying AA/GA rs361525 genotypes had a decreased risk for lung cancer compared to those with the GG genotype. Interestingly, the GG genotype displayed a tendency towards disease progression [54]. Another investigation by Liu N et al. indicated an increased risk of lung squamous cell carcinoma in carriers of the rs361525 GG genotype [52]. Zmorzyński, Sz. et al. found that the presence of rs361525 AA or AG genotypes correlated with elevated susceptibility to multiple myeloma (MM) and earlier onset of the disease, and patients diagnosed with MM predominantly exhibited the GG genotype [53]. The existing literature on the association between *TNF-α* rs361525 polymorphisms and breast cancer presents inconclusive results [52,53,54].

Currently, there is a lack of studies investigating the role of *TNF-α* polymorphisms in CLL susceptibility. Very few studies have been found that investigate these SNPs in hematological malignancies. Banescu et al. found no association with acute myeloid leukemia; however, rs1800750 was identified as an independent predictor for OS [13]. Another case–control study including 123 Brazilian patients with myeloproliferative neoplasms (MPN) revealed that the variant genotypes of rs361525 were significantly more prevalent in MPN cases (*p* = 0.04), representing a risk factor of 2.21 for developing MPN [56]. In another study, the rs1800750 heterozygous genotype was linked to a higher risk of Hodgkin’s lymphoma in Mexican patients (OR = 4.41, 95% CI: 1.21–16.6, *p* = 0.02) [57].

Similarly, in our investigation, the heterozygous genotype of rs1800750 was exclusively found in CLL (one case), and not in the control group, leading us to consider that it may be associated with an increased risk of CLL; however, we found very few patients carrying the variant A allele in both rs361525 and rs1800750, with no patients having the homozygous GG genotype, and only one patient having the heterozygous genotype on both loci. Furthermore, all these patients had other associated genetic risk factors (ex. unmutated *IGHV* gene) which have already been linked to CLL susceptibility. Although we cannot draw any conclusions on this matter, an interesting finding was that the one particular patient who presented the variant allele on the rs1800750 SNP carried several other genetic risk factors, as well, including an unmutated *IGHV* gene and a variant allele on all three of the other investigated SNPs (rs1800896, rs1800872, and rs361525).

The limitations of our study might be represented by the relatively small cohort of CLL patients, as well as the lack of data on the plasma levels of TNF-α and IL-10 in the patient samples and the lack of finance for comprehensive genetic analyses such as whole-exome/whole-genome analysis. However, to the best of our knowledge, our study is the first to investigate the role of *TNF-α* SNPs in CLL. To overcome some of our limitations, we included an analysis of additional genetic variants such as *IGHV* status, somatic mutations, DNA CNVs, and SNPs.

## 4. Materials and Methods

### 4.1. Patients

This research involved a cohort of 125 chronic lymphocytic leukemia (CLL) patients from the central region of Romania. At the time of their admission to the hematology departments in Târgu Mureș, peripheral blood samples were collected for analysis. These samples were obtained during the initial CLL diagnosis, prior to the commencement of chemotherapy. The inclusion criteria for the study required participants to have a confirmed diagnosis of CLL, be above 18 years of age, and provide signed informed consent to participate.

To obtain genomic DNA, Pure Link Genomic DNA kits from Thermo Fisher Scientific, located in Waltham, MA, USA, were utilized following the manufacturer’s instructions.

### 4.2. Analysis of the Mutational Status of the IGHV Gene

In this study, we initially performed PCR analysis using an *IGH* Somatic Hypermutation Assay v2.0 kit (available from Invivoscribe, located in San Diego, CA, USA), followed by separation of the amplified fragments by gel electrophoresis, to assess the somatic hypermutation status of the *IGHV* gene. This test utilizes Hypermutation Mix 1 and Hypermutation Mix 2 to amplify specific genomic DNA segments. The targeted segments lie between the upstream leader (VHL) or framework 1 (FR1) regions and the downstream joining (J) region of the *IGH* gene. Hypermutation Mix 1 focuses on amplifying sequences within the leader and joining regions, resulting in amplicon products that span the entire variable (V) region. The variable region encompasses the FR1, CDR1 (complementarity-determining region 1), FR2, CDR2, FR3, and CDR3 regions. On the other hand, Hypermutation Mix 2 targets sequences between the framework 1 (FR1) and joining (J) regions, generating amplicons that include a portion of the FR1 region extending to the downstream J region. The kit also contains Positive control (clonal) DNA, and Negative control (polyclonal/normal) DNA [58].

### 4.3. Testing for CNVs and Somatic Mutations

The presence of CNVs and gene mutations (*NOTCH1*, *SF3B1*, and *MYD88*) was tested by MLPA (multiplex ligation-dependent probe amplification), as previously described [44].

### 4.4. SNP Investigation

SNP genotyping was performed using TaqMan technology. Specifically, pre-designed TaqMan SNP Genotyping assays targeting *TNF-α* rs361525 and rs1800750, as well as *IL-10* rs1800896 and rs1800872, were utilized on a 7500 Fast Dx Real-time PCR System. A cohort of 239 healthy individuals, with similar age and gender distribution, were also tested for all SNPs for comparison.

The Ensembl genome browser, (release version 110, https://www.ensembl.org, accessed on 1 September 2023), was employed to assign wild-type and variant alleles to the studied SNPs [59].

### 4.5. Statistical Analysis

Statistical analyses were carried out using a Fisher Exact Test and a Mann–Whitney U Test, with the significance level of the *p* value being set to less than 0.05 in both cases. The free Haplotype Analysis software (version 1.05) was employed for the analysis of haplotypes involving the *IL-10* (rs1800896 and rs1800872) and *TNF-α* (rs361525 and rs1800750) SNPs. Kaplan–Meier curves were used to show the overall survival, and the log-rank test was used to compare the survival between the groups. To calculate hazard ratios (HR), the Cox proportional hazard model (Cox regression) was used. For all statistical analyses, GraphPad Prism 10.0.03 was used.

## 5. Conclusions

*IL-10* (rs1800896 and rs1800872) and *TNF-α* (rs361525 and rs1800750) SNPs and haplotypes are not associated with CLL risk. The absence of hypermutation in the *IGHV* gene has important prognostic value, being associated with short OS. Also, the individual risk factors for short OS were an age above 65 years at diagnosis and the presence of somatic mutations and/or CNVs. In the multivariable analysis, the presence of somatic mutations and the *IL-10* rs1800872 variant allele, and the association of CNVs with the *IL-10* rs1800896 variant allele, were identified as risk factors for short OS.

Moreover, the OS in unmutated *IGHV* patients was additionally affected (decreased) by the presence of CNVs and/or somatic mutations. Similarly, *Il-10* rs1800896 modulated the OS in unmutated *IGHV* patients with CNVs.

Comprehensive genetic investigation of CLL patients is crucial, because in most cases, there are several genetic risk factors present, and the association of multiple such alterations might have a significant influence on the prognosis.

## Figures and Tables

**Figure 1 ijms-25-01799-f001:**
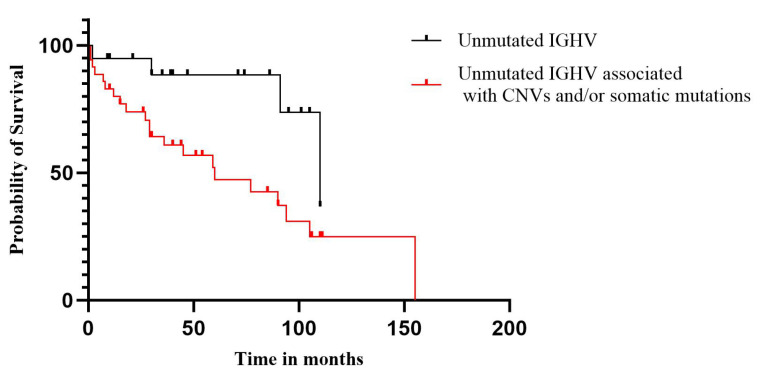
Survival of patients in whom the absence of SHM of the *IGHV* gene was associated with CNVs and/or somatic mutations.

**Figure 2 ijms-25-01799-f002:**
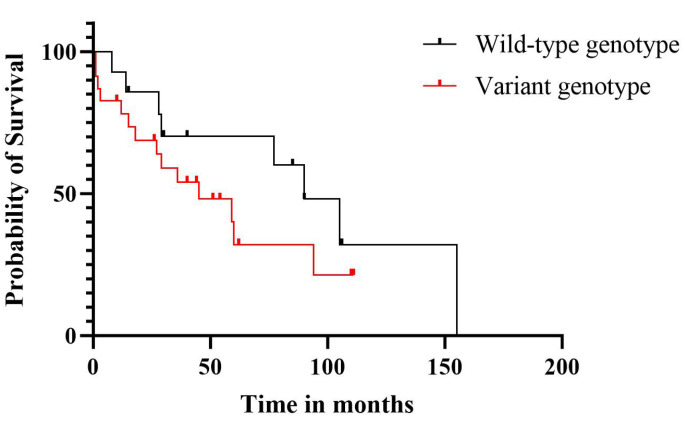
Survival of patients in whom the absence of SHM of the *IGHV* gene was associated with CNVs and rs1800896 (wild-type genotype: rs1800896 TT, unmutated *IGHV* and CNVs; variant genotype: rs1800896 TC + CC, unmutated *IGHV* and CNVs).

**Table 1 ijms-25-01799-t001:** Association of *IGHV* SHM with CNVs and/or somatic mutations.

		*IGHV*		*p* Value
		Mutated (%)	Unmutated (%)	
CLL with no CNVs and mutations		24 (19.2)	21 (16.8)	
CLL with CNVs *		40 (32.0)	11 (8.8)	0.009
	CLL with del13q **	14 (11.2)	5 (4.0)	0.67
	CLL with del11q **	3 (2.4)	0 (0.0)	-
	CLL with trisomy 12 **	5 (4.0)	0 (0.0)	-
	CLL with del17p **	2 (1.6)	0 (0.0)	-
CLL with somatic mutations ***		10 (8.0)	6 (4.8)	0.53
	CLL with *NOTCH1*	7 (5.6)	1 (0.8)	0.07
	CLL with *SF3B1*	3 (2.4)	5 (4.0)	0.41
CLL with somatic mutations ** and CNVs *		10(8.0)	1 (0.8)	0.02

CLL: chronic lymphocytic leukemia; * all identified CNVs taken together; ** the four most frequently identified CNVs; *** all investigated somatic mutations taken together. Data are presented as numbers (percentages); *p* values were obtained using the χ^2^ test.

**Table 2 ijms-25-01799-t002:** Distribution of genotypes in the CLL and control groups.

	Genotypes	CLL Group (% ^a^)	Control Group (% ^b^)	*p* Value
	Additive model			
*IL-10* rs1800896	TT	35.2	33.5	
	TC	50.4	47.7	0.97
	CC	14.4	18.8	0.43
	Dominant model			
	TC + CC	85.6	66.5	0.45
	Additive model			
*IL-10* rs1800872	GG	51.2	54.8	
	GT	40.8	39.7	0.73
	TT	8	5.4	0.36
	Dominant model			
	GT + TT	48.8	45.1	0.67
*TNF-α* rs361525	GG	96.8	95.6	0.70
	GA	3.2	4.4	
*TNF-α* rs1800750	GG	99.2	100	0.32
	GA	0.8	0	

CLL: chronic lymphocytic leukemia; ^a^ percentages were calculated for genotypes relative to the number of CLL cases; ^b^ percentages were calculated for genotypes relative to the number of controls. Data are presented as percentages; *p* values were obtained using the χ^2^ test.

**Table 3 ijms-25-01799-t003:** (**a**) Frequency of haplotypes in the two study groups for the *IL-10* SNPs (rs1800896 and rs1800872); (**b**) frequency of haplotypes in the two study groups for the *TNF-α* SNPs (rs361525 and rs1800750).

(a)				
Haplotype No.	Estimated Haplotypes for *IL-10* SNPs	Relative Frequencies in Controls	Relative Frequencies in CLL Patients	*p* Value
1	TTGG	0.13	0.10	
2	TTGT	0.16	0.17	0.53
3	TTTT	0.05	0.08	0.17
4	TCGG	0.24	0.27	0.43
5	TCGT	0.23	0.24	0.55
6	TCTT	0.01	0.00	1.00
7	CCGG	0.18	0.14	1.00
8	CCGT	0.01	0.00	1.00
**(b)**				
**Haplotype No.**	**Estimated Haplotypes for *TNF-α* SNPs**	**Relative Frequencies in Controls**	**Relative Frequencies in CLL Patients**	***p*** **Value**
1	GGGG	0.96	0.97	
2	GAGG	0.04	0.02	0.42
3	GAGA	0.00	0.01	0.27

## Data Availability

Additional data are available upon request.

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
