# Peer review of "Analysis of Mutational Status of IGHV, and Cytokine Polymorphisms as Prognostic Factors in Chronic Lymphocytic Leukemia: The Romanian Experience"

_ijms, 2024, doi:10.3390/ijms25031799_

Round 1

Reviewer 1 Report

Comments and Suggestions for Authors

In the manuscript, Balla and colleagues describe the mutational status of IGHV in relation with polymorphisms of cytokines IL-10 and TNF-alfa and their prognostic impact in Chronic Lymphocytic Leukemia. The study is quite well written, however it presents also some relevant issues. Overall, the scope of the manuscript is not clearly focused and Authors allegedly wish to report on the association between IGHV status and SNPs; however the manuscript presents very few data about this, namely the overall absence of significant associations between CLL and SNPs (the few differences in genotype composition are very quickly presented), a Cox regression and one Kaplan-Meyer curve.

A detailed point-by-point discussion is reported below.

-          The title reports rs736456 and rs3778076, but these SNPs related to TNF-a are not reported in the next

-          The introduction is excessively long and more appropriate to a review; Authors should focus on the most relevant aspects of cytokines, since also IGHV is an established marker with a decades-long history.

-          Overall, Authors define chromosomal aberrations as CNVs, without however stratifying the differential impact of, e.g., del17p and del13q. This issue needs to be addressed throughout the manuscript as also significantly impact the overall survival of CLL patients, later investigated.

-          Table 1 is somewhat confusing; are “CLL with somatic mutations” intended with multiple mutations? Was TP53 not investigated?

-          Figure 1 is quite difficult to interpret at first sight; Authors should add a title caption to each graph indicating IL-10 and TNF-alfa and provide a better legend: reporting a combined “CCTG” for the two SNPs is not of immediate interpretation for a non-geneticist; Authors may use either different colors for the two genotypes, or using a separator mark, or using a table with a title on the first row and then all the specific haplotype configurations, one in each column.

-          On figure 1: Authors discuss an enrichment of the TTGG configuration in the control group, with a delta of 3%; however, they do not discuss CCGG with a delta=4%.

-          Figure 2 does not provide significant data connected with the investigated topic of IGHV and cytokines SNPs. Authors should consider deleting or moving it to supplemental material.

-          In the Results section, supplementary materials but not present on the journal page.

-          Paragraph 2.4: I do not get the point. The impact of somatic mutations and chromosomal aberrations is already well-investigated and I don’t see any additional information in the data here reported. Moreover, Authors report a Cox regression (lines 278-279): is it a bivariate regression? A proper comparison should be against the univariate of somatic mutations only or CNV only, as the risk may come primarily from these lesions.

-          Figure 4: Are here all variant genotypes aggregated together? Authors also combine all CNVs indiscriminately, but there are radically different behaviors among chromosomal aberrations that should be taken into account. Are some of these genotypes associated with a particular lesion or there is a specific skewing between the two groups (even though all IGHV-UM and with genetic lesions)? If not, the difference here shown (derived from the additive effect of rs1800896 to unmutated IGHV/mutations) should come up more clearly beforehand.

Comments on the Quality of English Language

no specific comments

Author Response

responses attached 

Reviewer 2 Report

Comments and Suggestions for Authors

The authors analyzed the coincidence of IGHV mutational status and specific polymorphic alterations insome  cytokine genes in a Roumanian cohort of patients with chronic lymphocytic leukemia. The paper does not provide original or novel data of clinical relevance, but is of confirmatory nature.

Specific Points of Criticism:

(1)  The Title might be more „crisp“ or „interesting“ i f it would be shorter and conciser.

(2)  The Introduction is very extensive. Lines 51-188 have review character. On the other hand, while it is clearly a very lengthy introduction, it introduces well; maybe some repetitive passages may be omitted and other parts may be shortened.

(3)  Table 3 is very large and could go into the supplements.

(4)  Lines 329-361 are one single big text block andcould be split into several paragraphs.

Comments on the Quality of English Language

Minor English editing would be useful.

Author Response

responses atached
